# Generalizing graph matching beyond quadratic assignment model

**Tianshu Yu**
Arizona State University
tianshuy@asu.edu

**Junchi Yan**
Shanghai Jiao Tong University
yanjunchi@sjtu.edu.cn

**Yilin Wang**
Arizona State University
yilwang@adobe.com

**Wei Liu**
Tecent AI Lab
wl2223@columbia.edu

**Baoxin Li**
Arizona State University
baoxin.li@asu.edu

## Abstract

Graph matching has received persistent attention over several decades, which can be formulated as a quadratic assignment problem (QAP). We show that a large family of functions, which we define as Separable Functions, can approximate discrete graph matching in the continuous domain asymptotically by varying the approximation controlling parameters. We also study the properties of global optimality and devise convex/concave-preserving extensions to the widely used Lawler's QAP form. Our theoretical findings show the potential for deriving new algorithms and techniques for graph matching. We deliver solvers based on two specific instances of Separable Functions, and the state-of-the-art performance of our method is verified on popular benchmarks.

## 1 Introduction

Given two graphs, graph matching algorithms (GM) seek to find node-to-node correspondences by optimizing a pre-defined affinity score function. This problem falls into the category of quadratic assignment problem (QAP) [1], and has wide applications from object categorization [2] to protein alignment [3]. While a line of works using combinatorial heuristics [4, 5] attempt to solve graph matching, relaxation of original problem into the continuous domain is mostly employed and solved with different optimization techniques e.g. gradient [6] or multiplication [7, 8] based methods. The dominance of continuous relaxation may be partly because it is easier to analyze the local behavior of continuous functions, and one can often find a local optimum. In this paper, we focus on continuous relaxation of graph matching.

Graph matching seeks the solution to the quadratic assignment problem $\max_{\mathbf{X}} \text{vec}(\mathbf{X})^\top \mathbf{A} \text{vec}(\mathbf{X})$, where $\text{vec}(\mathbf{X}) \in \{0, 1\}^{n^2}$ is the column-wise vectorized version of the binary (partial) assignment matrix $\mathbf{X} \in \{0, 1\}^{n \times n}$ and the so-called affinity matrix $\mathbf{A} \in \Re^{n^2 \times n^2}$ in the real domain consists of the affinity score measuring how one edge in one graph is similar to another from the other graph. Traditionally, the common practice is relaxing $\text{vec}(\mathbf{X})$ into the continuous real domain $\text{vec}(\mathbf{X}) \in \Re^{n^2}$ [9, 7, 10].

In this paper, we show that a large family of functions, defined as Separable Functions, can asymptotically approximate the discrete matching problem by varying the approximation controlling parameters. With this function family, there exist infinite modelings of graph matching problem, thereby providing the feasibility of adapting different practical problems with different models. This provides a new perspective of considering graph matching. We also give analysis on the conditions based on which

these approximations have good properties. Novel solvers on instances of Separable Functions are proposed based on the path-following and multiplicative techniques respectively.

**Notations** We use bold lower-case $\mathbf{x}$ and upper-case $\mathbf{A}$ to represent vector and matrix, respectively. Function $\mathrm{vec}(\cdot)$ transforms a matrix to its column-wise vectorized replica. Conversely, function $\mathrm{mat}(\cdot)$ transfers a vector back to its matrix form. Denote $\Re_+$, $\mathbb{S}$ as non-negative real numbers and symmetric matrices respectively. Function $\mathbf{K} = \mathrm{diag}(\mathbf{k})$ transforms a vector $\mathbf{k}$ into a diagonal matrix $\mathbf{K}$ such that $\mathbf{K}_{ij} = \mathbf{k}_i$ if $i = j$, and $\mathbf{K}_{ij} = 0$ otherwise.

## 2 Related Work

Different from linear assignment [11], the quadratic assignment problem (QAP) in terms of graph matching in the literature is often formulated in two forms: i) the Koopmans-Beckmann's QAP [12]: $\mathrm{tr}(\mathbf{X}^\top \mathbf{A}_i \mathbf{X} \mathbf{A}_j) + \mathrm{tr}(\mathbf{A}_p^\top \mathbf{X})$ where $\mathbf{X}$ is the assignment matrix, $\mathbf{A}_i$ and $\mathbf{A}_j$ are the weighted adjacency matrices, and $\mathbf{A}_p$ is the node-to-node similarity matrix. Methods based on this formula include [13, 14, 15], to name a few; ii) the more general Lawler's QAP [16] by $\mathrm{vec}(\mathbf{X})^\top \mathbf{A} \mathrm{vec}(\mathbf{X})$. Note that the Koopmans-Beckmann's QAP can always be represented as a special case of the Lawler's by setting $\mathbf{A} = \mathbf{A}_j \otimes \mathbf{A}_i$, and many previous works [9, 10, 6, 17] adopt the Lawler's form, which is also the main focus of this paper for its generality. A recent survey [18] provides a more comprehensive literature review.

Though there are a few (quasi-)discrete methods [5, 19, 20] that directly work in the binary domain, the major line of research falls into the following tracks in the continuous domain. Our relaxation techniques do not fall into any of these categories and opens up new possibility for new algorithms.

**Spectral relaxation**: The authors of the seminal work in [7] proposed to relax $\mathbf{X}$ to be of unit length $\|\mathrm{vec}(\mathbf{X})\|_2^2 = 1$, and the resulting optimization problem can be efficiently solved by computing the leading eigen-vector of the affinity matrix $\mathbf{A}$. Better approximation has been made in [21] by adding an affine constraint. In contrast to the above Lawler's QAP based models, there are a few earlier methods [13, 22] based on the Koopmans-Beckmann's QAP, and the relxation is often fulfilled by setting $\mathbf{X}^\top \mathbf{X} = \mathbf{I}$, where $\mathbf{I}$ is the identify matrix. In general, spectral relaxation is efficient while not tight, which hinders the matching accuracy.

**Semi-definite programming relaxation**: SDP has been a standard tool for combinatorial problem, and it has been adopted to tackle the graph matching problem. In existing work, a variable $\mathbf{Y}$ subject to $\mathbf{Y} = \mathrm{vec}(\mathbf{X})\mathrm{vec}(\mathbf{X})^\top$ is introduced. As a result, the raw problem is approximated by SDP which relaxes the non-convex constraint on $\mathbf{Y}$ into a semi-definite one: $\mathbf{Y} \succeq \mathrm{vec}(\mathbf{X})\mathrm{vec}(\mathbf{X})^\top$. The final matching $\mathbf{X}$ is then recovered by different heuristics such as winner-take-all [23] or randomization [24]. However, the SDP solver is not very popular in graph matching, as the variable $\mathbf{Y}$ squares the raw variable size, resulting in high complexity.

**Doubly-stochastic relaxation**: Based on the fact that doubly-stochastic matrix is the convex hull of the permutation matrix, various methods have been proposed in this line to formulate the relaxed problem into a non-convex quadratic programming problem for both Koopmans-Beckmann's and Lawler's QAP. Linear programming is adopted in [25] to approximate the quadratic problem, followed by more complex path following methods [14, 15] to approximate the relaxed quadratic problem – all are based on the Koopmans-Beckmann's QAP. For the more general Lawler's QAP, the seminal work termed graduated assignment [9] approximates the relaxed QAP via solving a series of linear approximations via iterative Taylor expansions. A random walk perspective to the graph matching problem is adopted in [10], whereby the method can also be seen as a weighting solution by [9] and the multiplication method [7]. More recently factorized graph matching is devised in [17], which also follows the doubly-stochastic relaxation on top of other relaxations on the objective function.

Finally we also briefly review recent advances on hyper and multiple graph matching. There are studies addressing the more general hypergraph matching problem, whereby the third-order or higher is considered in the objective and usually an affinity tensor is adopted. Many current hypergraph matching methods approximate the third-order objective via iterative approximations. In each iteration, often a Lawler's QAP is involved [26, 27]. Moreover, the Lawler's QAP model solvers can also be used in matching a batch of graphs beyond two graphs. In [28], an alternating optimization method was proposed, whereby in each iteration a Lawler's QAP problem is derived and solved. [29] further extends multi-graph matching problem to an online version. These connections further

highlight the importance of the Lawler's QAP for not only traditional graph matching, but also hypergraph matching and multiple graph matching.

## 3 Generalizing the QAP for Graph Matching

We re-visit the graph matching problem in this section. We propose an equivalent model to the discrete one over continuous domain $[0, 1]$, provided the relaxation gap is 0. This gives rise to the possibility to relax graph matching with much tighter ways. Mathematically, graph matching can be formulated as the following quadratic assignment problem which is also called Lawler's QAP[1] [16]:

$$\max_{\mathbf{X}} \text{vec}(\mathbf{X})^{\top} \mathbf{A} \text{vec}(\mathbf{X})$$
$$\text{s.t. } \mathbf{X}\mathbf{1} = \mathbf{1}, \mathbf{X}^{\top}\mathbf{1} = \mathbf{1}, \mathbf{x}_{ia} \in \{0, 1\} \quad (1)$$

where $\mathbf{A} \in \Re_{+}^{n^2 \times n^2}$ is a non-negative affinity matrix, which encodes node similarities on diagonal elements and edge similarities on the rest. Note $\mathbf{x}_{ia}$ denotes the element of $\mathbf{X}$ indexed by row $i$ and column $a$ indicating the matching status of node $i$ to node $a$ from the other graph. If we break down problem (1) into element-wise product, it becomes:

$$\max_{\mathbf{x}} \sum_{i,j,a,b} \mathbf{A}_{ij:ab}\mathbf{x}_{ia}\mathbf{x}_{jb}$$
$$\text{s.t. } \mathbf{H}\mathbf{x} = \mathbf{1}, \mathbf{x} \in \{0, 1\}^{n^2} \quad (2)$$

where $\mathbf{A}_{ij:ab}$ corresponds to the edge similarity between edge $(i, j) \in \mathcal{G}_1$ and $(a, b) \in \mathcal{G}_2$. Here $\mathbf{H} \in \{0, 1\}^{2n \times n^2}$ is a selection matrix over the elements of $\mathbf{x}$ sufficing assignment constraints according to (1).

In particular, we relax $\mathbf{x}$ into continuous domain and let $f_{\text{prod}}(\mathbf{x}_{ia}, \mathbf{x}_{jb}) = \mathbf{x}_{ia}\mathbf{x}_{jb}$:

$$\max_{\mathbf{x}} \sum_{i,j,a,b} \mathbf{A}_{ij:ab}f_{\text{prod}}(\mathbf{x}_{ia}, \mathbf{x}_{jb})$$
$$\text{s.t. } \mathbf{H}\mathbf{x} = \mathbf{1}, \mathbf{x} \in [0, 1]^{n^2} \quad (3)$$

We generalize problem (3) by replacing $f_{\text{prod}}$ with $f_{\delta}$:

$$\max_{\mathbf{x}} \sum_{i,j,a,b} \mathbf{A}_{ij:ab}f_{\delta}(\mathbf{x}_{ia}, \mathbf{x}_{jb})$$
$$\text{s.t. } \mathbf{H}\mathbf{x} = \mathbf{1}, \mathbf{x} \in [0, 1]^{n^2} \quad (4)$$

where $f_{\delta}$ is a 2D quasi-delta function in the continuous domain ($f_{\delta}(x, y) = 1$ if $x = 1$ and $y = 1$, and $f_{\delta} = 0$ otherwise). We have the following theorem that establishes the connection between (2) and (4):

**Theorem 1** *The optimal objective $p^*$ to problem (2) is equal to the optimal objective $p_{\delta}^*$ to problem (4).*

**Remark** Based on Theorem 1, one can devise a sampling procedure to find the optimal solution to problem (2) from the solution to problem (4): Given optimal $\mathbf{x}_{\delta}^*$ to problem (4), if all the elements are in the set $\{0, 1\}$, then $\mathbf{x}_{\delta}^*$ is automatically optimal to problem (2). If not, we first remove all 1 elements and corresponding columns and rows, yielding a subvector (submatrix) $\mathbf{x}^{\dagger}$ with all elements in range $[0, 1)$. Then any sampling subject to one-to-one constraint on $\mathbf{x}^{\dagger}$, together with the removed discrete values, forms the optimal solution to problem (2).

For the time being, a discrete assignment problem (2) is relaxed into (4) with continuous feasible domain. However, function $f_{\delta}$ is not continuous as there is a jump at value $(1, 1)$, ending up with much difficulty to solve (recall (4) is equivalent to (2)). In the next section, we will show some approximate techniques to tackle problem (4).

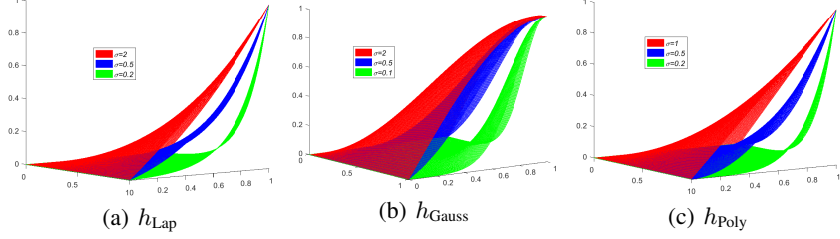

| | | |
|---|---|---|
| (a) $h_{\mathrm{Lap}}$ | (b) $h_{\mathrm{Gauss}}$ | (c) $h_{\mathrm{Poly}}$ |

Figure 1: Three examples of approximations (Laplacian, Gaussian, Polynomial) to function $f_\delta$ with varying $\theta$. The closer for $\theta \to 0$ (from red to green), the better approximation to $f_\delta$.

## 4 Separable Functions

### 4.1 Separable Approximation Function Family

It is important to find an appropriate approximate function for $f_\delta$, otherwise it may lead to intractable models to solve. To avoid high computational cost, we narrow our focus on a specific family of functions, called Separable Functions.

**Definition 1** *A function $f_\theta(x, y)$ is called Separable Function (**SF**)[2] if it satisfies the following properties:*

*1. $f_\theta(x, y) = h_\theta(x) \times h_\theta(y)$ where $h_\theta$ is defined on $[0, 1]$.*

*2. $h_\theta(0) = 0$ and $h_\theta(1) = 1$. $h_\theta \in \mathbb{C}^1$.*

*3. $h_\theta$ is non-decreasing and $\lim_{\theta \to 0} h_\theta(x) - h_\delta(x) = 0$ for any $x \in [0, 1]$, where $h_\delta$ is defined on $[0, 1]$, $h_\delta(x) = 1$ if $x = 1$ and $h_\delta(x) = 0$ otherwise.*

We also call such a function $h_\theta$ **univariate SF**, where $\theta$ is a controlling parameter. Being seemingly a simple formulation, SF has three fine properties for computation.

Firstly, SF shows similar behavior as a probabilistic distribution on two independent variables. That is, if two nodes are impossible to match, then any pair of edges containing the two nodes will never match neither. Mathematically, assuming the matching score of node pair $\langle i, a \rangle$ is $h_\theta(\mathbf{x}_{ia})$, we have $f_\theta(\mathbf{x}_{ia}, \mathbf{x}_{jb}) = 0$ for any $\langle j, b \rangle$ if $h_\theta(\mathbf{x}_{ia}) = 0$.

Secondly, the definition of SF eases gradient computing. For a given SF $f_\theta(x, y) = h_\theta(x)h_\theta(y)$, the approximate version of problem (4) can be expressed in matrix form as:

$$\max_{\mathbf{x}} \mathbf{h}_\theta^\top \mathbf{A} \mathbf{h}_\theta$$
$$\text{s.t. } \mathbf{H}\mathbf{x} = \mathbf{1}, \mathbf{x} \in [0, 1]^{n^2} \tag{5}$$

where $\mathbf{h}_\theta = [h_\theta(\mathbf{x}_1), ..., h_\theta(\mathbf{x}_{n^2})]^\top$. The gradient of objective (5) with respect to $\mathbf{x}$ is $\nabla \mathbf{x} = 2\mathbf{G}\mathbf{A}\mathbf{h}$, where $\mathbf{G}$ is a diagonal matrix with the $i$th element $\partial h_\theta(\mathbf{x}_i)/\partial \mathbf{x}_i$.

The third advantage of SF is that we can construct new approximation functions via reweighted summation and multiplication of existing ones, e.g. if $h_1$ and $h_2$ are two univariate SFs, it can be trivially verified that $\alpha h_1 + (1 - \alpha)h_2$ for $0 \le \alpha \le 1$ and $h_1 \times h_2$ are also univariate SFs.

If we keep the constraints on $\mathbf{x}$ intact as in problem (5), and let $p_\theta^* = \max_\mathbf{x} \mathbf{h}_\theta(\mathbf{x})^\top \mathbf{A} \mathbf{h}_\theta(\mathbf{x})$, where $\mathbf{h}_\theta(\mathbf{x}) = [h_\theta(\mathbf{x}_1), ..., h_\theta(\mathbf{x_n})]^\top$, we have the following theorem:

**Theorem 2** $\lim_{\theta \to 0} p_\theta^* = p_\delta^*$

See supplementary material for proof details. The above theorem guarantees that, if we approximate the quasi-delta function by letting $\theta \to 0$, problem (4) can also be approximated asymptotically. As $h_\theta \in \mathbb{C}^1$, gradient-base algorithms can be applied to such approximations.

## 4.2 Approximations to Function $f_\delta$

Though we have proved that using $f_\delta$ can derive an equivalent problem i.e. (4), finding its optimal solution is still notoriously difficult. Instead of solving (4) directly, based on the analysis in Sec. 4.1, we introduce approximation functions to $f_\delta$. To simplify the expression, we only present the univariate SF $h$, and the SF $f$ can be obtained using definition (1). It is trivial to show that the SFs derived from the following functions approximate $f_\delta$ when $\theta \to 0_+$ under the properties in definition (1):

$$h_{\text{Lap}}(x) = \frac{1}{m} \left\{ \exp\left(\frac{x-1}{\theta}\right) - d \right\} \tag{6a}$$

$$h_{\text{Gauss}}(x) = \frac{1}{m} \left\{ \exp\left(-\frac{(x-1)^2}{\theta}\right) - d \right\} \tag{6b}$$

$$h_{\text{Poly}}(x) = x^{\frac{1}{\theta}} \tag{6c}$$

where $d = \exp(-\frac{1}{\theta})$ and $m = 1 - d$. The usage of $m$ and $d$ is to normalize the SFs to satisfy the second property. Figure 1 shows some examples of such functions with varying $\theta$ values. Note that traditional quadratic graph matching model in fact is a special case of our model, which seeks to optimize a model where the SF is derived from $h_{\text{Poly}}$ and $\theta = 1$. Specifically, for the univariate SFs (6a) and (6c), we also have the following proposition.

**Proposition 1** *For univariate SF $h_{Lap}$, $h_{Poly}$, suppose $p_1^*$ and $p_2^*$ are the optimal objectives for (5) with $\theta_1$ and $\theta_2$, respectively. Then we have $p_1^* \geq p_2^*$ if $0 < \theta_2 < \theta_1$.*

Together with Theorem 2, this claim means that, given univariate SF $h_{\text{Lap}}$ or $h_{\text{Poly}}$, the optimal objective of (5) will converge as $\theta \to 0_+$ monotonically.

## 4.3 Convexity/Concavity Analysis

Section 4.1 and 4.2 show that original problem (4) can be asymptotically approximated using SFs as $\theta \to 0$. In this section, we analyze the properties of convexity/concavity under such approximations. We believe this effort is worthwhile as one can employ techniques e.g. self-amplification [30], to convert non-convex/non-concave problems into convex/concave ones with the beneficial properties of convexity/concavity. We first show the equivalence of problem (3) and (5) under global convexity.

**Theorem 3** *Assume that affinity $\mathbf{A}$ is positive definite. If the univariate SF $h_\theta(x) \leq x$ on $[0,1]$, then the global maxima of problem (2), which is discrete, must also be the global maxima of problem (5).*

The above theorem builds up a link from problem (2) to problem (5) when $\mathbf{A}$ is positive definite. In this case, we first conclude that the optimum to problem (3) is discrete, hence also optimal to (2). Then as long as $h_\theta(x) < x$ on $[0,1]$ and $h_\theta$ satisfies the second property in Definition 1, this solution is also optimal to problem (5). In this case the optimal objective gap of these three problems becomes $0$. We give the following proposition showing under mild conditions, the generalized problem (5) is convexity/concavity-preserving.

**Proposition 2** *Assume affinity maxtrix $\mathbf{A}$ is positive/negative semi-definite, then as long as the univariate SF $h_\theta$ is convex, the objective of (5) is convex/concave.*

Any matrix $\mathbf{A}$ can be transformed to positive definite by adding up a diagonal matrix $\lambda \mathbf{I}$. The lower bound of $\lambda$ is $\lambda \geq |\lambda^\dagger|$, where $\lambda^\dagger$ is the smallest eigenvalue of $\mathbf{A}$ below 0. We define $\mathbf{A}^\dagger = \mathbf{A} + \lambda \mathbf{I}$.

**Proposition 3** *Assume affinity matrix $\mathbf{A}$ is positive definite and univariate SF $h_\theta$ is convex. The optimal value to the following problem is:*

$$E_{conv} = \max_{\mathbf{x}} \mathbf{h}_\theta^\top \mathbf{A}^\dagger \mathbf{h}_\theta \tag{7}$$

*Then there exists a permutation $\mathbf{x}^*$, s.t. $E_{conv} - E(\mathbf{x}^*) \leq n\lambda$ where $E(\mathbf{x}^*)$ is the objective value w.r.t. problem (5).*

# 5 Two Optimization Strategies for Generalized GM

---
**Algorithm 1** Path following for GGM
---
    **Input: A**, $h_\theta$, $\theta_0$, $0 < \alpha < 1$, initial $\mathbf{x}_0$, $k$;
    **Output: x**
    $\mathbf{x} \leftarrow \mathbf{x}_0$, $\theta \leftarrow \theta_0$
    **repeat**
        make problem according to (5) with $\theta$
        **repeat**
            compute **V** using formula (8)
            $\mathbf{x} = \mathbf{x} + \epsilon\mathrm{vec}(\mathbf{V})$
        **until** Converge
        $\theta \leftarrow \alpha\theta$
    **until** $\theta < k$

---
**Algorithm 2** Multiplicative strategy for GGM
---
    **Input: A**, $h_\theta$, initial $\mathbf{x}_0$; **Output: x**
    $\mathbf{x} \leftarrow \mathbf{x}_0$
    **repeat**
        $\mathbf{h} \leftarrow h_\theta(\mathbf{x})$
        $\mathbf{h} \leftarrow \mathbf{A}\mathbf{h}$
        $\mathbf{x} \leftarrow h_\theta^{-1}(\mathbf{h})$
    **until** Converge
---

## 5.1 Path Following Strategy

It is observed that solving the problem when $\theta$ is too close to $0$ is highly non-convex, suggesting the existence of many local optima. Instead, moderate smoothness is desired when we initiate the optimization. This naturally leads to the path following strategy. Such optimization is involved in [9, 17, 31]. In our implementation, we start by obtaining a local optimum $\mathbf{x}_1^*$ from a relatively tractable problem $\mathcal{P}_{\theta_1}$, then we shrink the value of $\theta_1$ by letting $\theta_2 = \alpha\theta_1$ where $0 < \alpha < 1$. Let the starting point for next iteration be $\mathbf{x}_1^*$, we solve the updated problem $\mathcal{P}_{\theta_2}$. The iteration continues until convergence condition is satisfied. To verify the convergence, we calculate the energy gap between two consecutive iterations. Formally, for current $\mathbf{x}^{(t)}$ at iteration $t$, we calculate the corresponding energy $\mathcal{E}^{(t)} = \mathbf{x}^{(t)\top}\mathbf{A}\mathbf{x}^{(t)}$. The energy at previous iteration $t-1$ is analogously calculated as $\mathcal{E}^{(t-1)} = \mathbf{x}^{(t-1)\top}\mathbf{A}\mathbf{x}^{(t-1)}$. Then if $\left|\mathcal{E}^{(t)} - \mathcal{E}^{(t-1)}\right| < \eta$, where $\eta$ is a small positive value, we identify the convergence of the iteration. If there is no such $t$, the algorithm stops when reaching the pre-defined maximal iteration number. In all the following experiments, we let $\eta = 10^{-8}$.

Note the problem $\mathcal{P}_\theta$ is a general objective with affine constraints. For any gradient-based strategy, projection is necessary to mapping the current solution back to the feasible set. As discussed in [8], projection in variable domain may lead to weak optima. Instead, we use Iterative Bregmann Gradient Projection (IBGP), which is performed in the gradient domain and the convergence is guaranteed [32]. Given current gradient $\mathbf{U} = \mathrm{mat}(\nabla\mathbf{x})$, previous matching $\mathbf{X}$ and step length $\epsilon$, IBGP performs the following calculations iteratively to obtain $\mathbf{V}$ until convergence:

$$\mathbf{V} = \mathbf{U} - \frac{1}{n}\mathbf{U}\mathbf{1}\mathbf{1}^T - \frac{1}{n}\mathbf{1}\mathbf{1}^\top\mathbf{U} + \frac{2}{n^2}\mathbf{1}\mathbf{1}^\top\mathbf{U}\mathbf{1}\mathbf{1}^\top \tag{8a}$$

$$\mathbf{V}_{ij} = -\mathbf{X}_{ij}/\epsilon \quad \text{if} \quad \mathbf{V}_{ij} < -\mathbf{X}_{ij}/\epsilon \tag{8b}$$

$$\mathbf{V}_{ij} = (1 - \mathbf{X}_{ij})/\epsilon \quad \text{if} \quad \mathbf{V}_{ij} > (1 - \mathbf{X}_{ij})/\epsilon \tag{8c}$$

where $\mathbf{V}$ is the update direction within the feasible set. Note the iterative procedure in the above equation is a projection. As the constraint set is convex (affinity set), the projection convergence is ensured. Thus in each iteration of update, the algorithm seeks a direction $\mathbf{V}$ with ascending guarantee and proceeds a fixed length $\epsilon$. This procedure iterates until convergence or maximal step number. The path following method is summarized in Algorithm 1.

## 5.2 Multiplication Strategy

Multiplicative strategy on optimizing quadratic objective proved to be convergent under the assumption that $\mathbf{A}$ is positive semi-definite [33]. In this strategy, each step amounts to a multiplication $\mathbf{x}^{(t+1)} = \mathbf{A}\mathbf{x}^{(t)}$ and the objective score over the solution path is non-decreasing. There are works [10, 9, 6] falling into this category. However, in general affinity $\mathbf{A}$ is barely positive semi-definite. While some methods handle this circumstance by adding reweighted identity matrix to $\mathbf{A}$ [34], others simply neglect the non-decreasing constraint including some popular algorithms [10, 9]. The empirical success of such methods suggests pursuing objective ascending and enhancing matching

accuracy sometimes are paradox. Moreover, the recent study [35] further shows due to noise and the parametric modeling limitation of the affinity function, high accuracy may even corresponds to lower affinity score. Inspired by these observations, we devise a simple yet effective multiplicative strategy by omitting the non-decreasing check. The procedure is shown in Algorithm 2. In this strategy, the update rule involves calculating inverse function of $h_\theta$. While it is found the multiplicative method converges much faster and hence the overall run time is less compared with the path following method.

## 6 Experiments

Three popular benchmarks are used including Random Graph Matching [10], CMU house sequence [36] and Caltech-101/MSRC object matching [10]. *accuracy*, *score* and *ratio* are evaluated, where *accuracy* measures the portion of correctly matched nodes with respect to all nodes, *score* represents the value of the objective function and *ratio* emphasizes the ratio between current objective value and the maximal one. The algorithms for comparison include Spectral Matching (**SM**) [7], Integer Projected Fixed Point (**IPFP**) [19], Graduated Assignment (**GAGM**) [9], Reweighted Random Walk (**RRWM**) [10], Soft-restricted Graduated Assignment (**SRGA**) [6], Factorized Graph Matching (**FGM**) [17] and Branching Path Following Matching (**BPM**) [31]. We term our algorithm Generalized Graph Matching (**GGM**) with a subscript indicating the corresponding Separable Function and optimization strategy. Namely, **GGM**$_{xy}$ represents the method with Separable Function $x \in \{l : h_{\text{Lap}}; p : h_{\text{Poly}}\}$ and optimizing strategy $y \in \{p : \text{Path following}; m : \text{Multiplication}\}$. In all the experiments, the algorithms with any updating rules are initialized with a uniform matching. For path following strategy of GGM, we set $\theta_0 = 2$, $\alpha = 0.5$, $k = 0.2$.

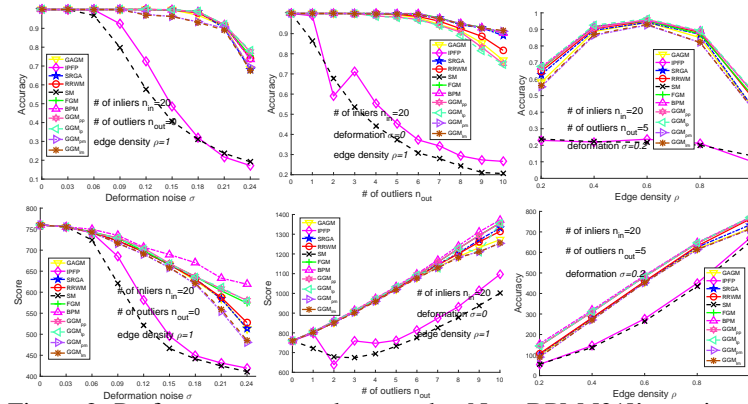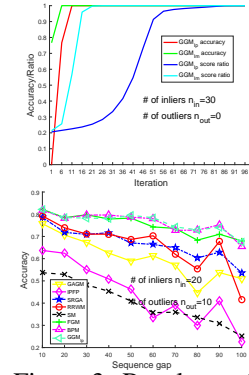

Figure 2: Performance on random graphs. Note BPM [31]'s runtime is significantly more expensive than other methods (empirically an order higher than ours using the public source code) as it simultaneously seeks multiple paths for the best score (though accuracy is similar to ours). In contrast, our method focus on one path no matter the path following or multiplicative strategy is used.

Figure 3: Results on C-MU house. **Upper:** convergence speed vs iteration. **Lower:** accuracy by frame gap.

**Random Graph Matching** This test is performed following the protocol in [10]. For each trial, source graph $\mathcal{G}_S$ and destination graph $\mathcal{G}_D$ with $n_{in}$ inlier nodes are generated, consisting of vector attributes $\mathbf{a}_{ij}^S$ and $\mathbf{a}_{ij}^D$ for both nodes and edges (note $\mathbf{a}_{ii}$ is a node attribute and $\mathbf{a}_{ij}$ is an edge attribute when $i \neq j$.). In the initial setting, $\mathcal{G}_D$ is the replica of $\mathcal{G}_S$. Three types of sub-experiments are conducted with varying graph deformation $\sigma$, number of outliers $n_{out}$ and edge density $\rho$. To deform a graph, we add an independent Gaussian noise $\varepsilon \sim \mathcal{N}(0, \sigma)$ to attribute $\mathbf{a}_{ij}^D$ such that $\mathbf{a}_{ij}^D = \mathbf{a}_{ij}^S + \varepsilon$. Thus the resulting affinity is calculated by $\mathbf{A}_{ij:ab} = \exp(-|\mathbf{a}_{ij}^S - \mathbf{a}_{ab}^D|^2 / \sigma_s^2)$. The parameter $\sigma_s$ is empirically set to be $0.15$. In outlier test, we generate the same number of outlier nodes for both graph. In edge density test, we randomly sample $\rho$ portion of corresponding edges from two fully connected graphs. Each type of sub-experiment is independently carried out 500 times, while average *accuracy* and *score* are calculated.

Results are shown in Fig 2. In the deformation and the edge density tests, GGM$_{pp}$ and GGM$_{lp}$ achieve competitive performance compared to state-of-the-art algorithms. Especially when there

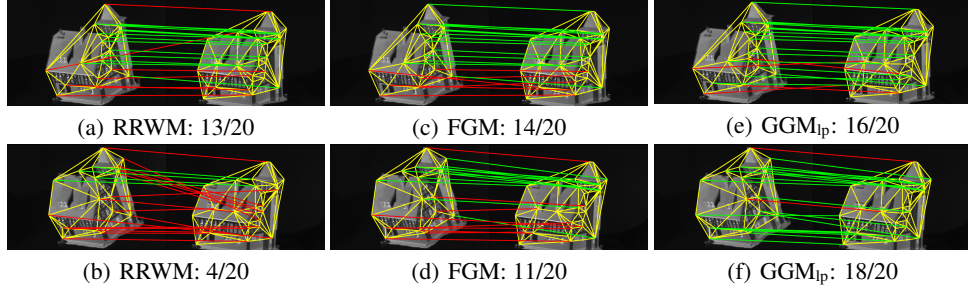

| (a) RRWM: 13/20 | (c) FGM: 14/20 | (e) GGM$_{\mathrm{lp}}$: 16/20 |
|---|---|---|
| (b) RRWM: 4/20 | (d) FGM: 11/20 | (f) GGM$_{\mathrm{lp}}$: 18/20 |

Figure 4: Top and bottom row shows examples on CMU house sequence with gap $20$ and $80$ respectively, by setting $(n^S = 30, n^D = 20)$.

is combination of severe deformation and edge density is low, GGM$_{pp}$ and GGM$_{lp}$ outperform the selected counterparts. On the other hand, GGM$_{pm}$ and GGM$_{lm}$ reach significant performance close to state-of-the-art e.g. BPM [31]. Though multiplicative strategies cannot guarantee ascending objective in each iteration, GGM$_{pm}$ and GGM$_{lm}$ are still effective. This supports the discussion of the paradox between matching accuracy and objective score in Section 5.2. We only show results of GGM$_{\mathrm{lp}}$ in the following experiments, as we see no notable performance gap compared to the other settings.

To examine the algorithm sensitivity to the parameters, we also conduct an extra Random Graph Matching experiment with SFs SFs $h_{\mathrm{Poly}}$ and $h_{\mathrm{Lap}}$ on Algorithm 1. In this test, we let deformation noise $0.15$ and edge density $0.8$, 20 inliers and 5 outliers. Test is carried out independently for 20 times and the average accuracy is reported. For both the SFs, we observe that $k = 0.2$ is sufficient to produce satisfying matching accuracy. Thus we conduct the test by varying the values of $\theta_0$ and $\alpha$. The results are demonstrated in Table 1. As larger $\theta_0$ and $\alpha$ indicate more iterations, and $\theta_0 < 2$ and $\alpha < 0.5$ result in decreasing behavior, we employ the setting $\theta_0 = 2$ and $\alpha = 0.5$ throughout all experiments.

Table 1: Sensitivity test on $h_{\mathrm{Poly}}$ and $h_{\mathrm{Lap}}$

| $h_{\mathrm{Poly}}$ | | $\alpha$ | | | | |
|---|---|---|---|---|---|---|
| | | 0.7 | 0.6 | 0.5 | 0.4 | 0.3 |
| $\theta_0$ | 3 | 0.842 | 0.839 | 0.841 | 0.721 | 0.610 |
| | 2 | 0.905 | 0.905 | 0.904 | 0.848 | 0.725 |
| | 1 | 0.910 | 0.905 | 0.908 | 0.851 | 0.717 |
| | 0.5 | 0.823 | 0.814 | 0.770 | 0.652 | 0.422 |

| $h_{\mathrm{Lap}}$ | | $\alpha$ | | | | |
|---|---|---|---|---|---|---|
| | | 0.7 | 0.6 | 0.5 | 0.4 | 0.3 |
| $\theta_0$ | 4 | 0.912 | 0.909 | 0.910 | 0.872 | 0.685 |
| | 3 | 0.911 | 0.907 | 0.903 | 0.836 | 0.672 |
| | 2 | 0.904 | 0.904 | 0.906 | 0.811 | 0.567 |
| | 1 | 0.853 | 0.844 | 0.810 | 0.728 | 0.472 |

**CMU House Sequence** We perform feature point matching on widely used CMU house sequence dataset following the settings in [36, 10]. The dataset consists of 111 house images with gradually changing view points. There are 30 landmark points in each frame. Following the protocol in [10, 31], matching test is conducted on totally 560 pairs of images, spaced by varying frame gaps $(10, 20, ..., 100)$. We use 2 settings of nodes $(n^S, n^D) = (30, 30)$ and $(20, 30)$. In case $n^S < 30$,

Table 2: Performance on natural images from Caltech-101 and MSRC dataset.

| Method | GAGM | IPFP | SRGA | RRWM | SM | FGM | BPM | GGM$_{\mathrm{lp}}$ |
|---|---|---|---|---|---|---|---|---|
| *accuracy (%)* | 73.66 | 75.77 | 72.86 | 72.95 | 65.78 | 76.35 | 75.14 | **76.69** |
| *score ratio* | 0.933 | 0.942 | 0.940 | 0.946 | 0.735 | 0.969 | 1 | 0.972 |

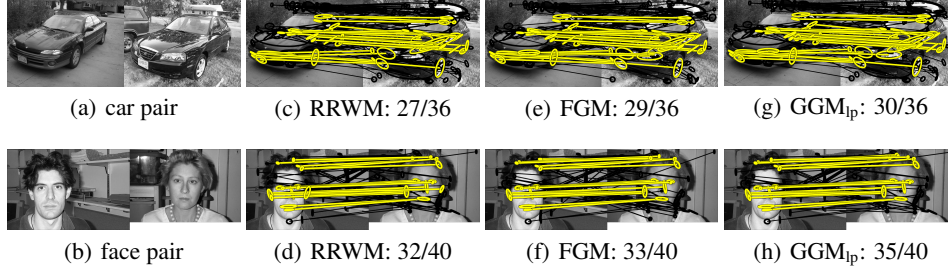

(a) car pair     (c) RRWM: 27/36     (e) FGM: 29/36     (g) GGM$_{\text{lp}}$: 30/36

(b) face pair    (d) RRWM: 32/40     (f) FGM: 33/40     (h) GGM$_{\text{lp}}$: 35/40

Figure 5: Examples of matchings on selected Caltech-101 and MSRC dataset.

$n^S$ nodes are randomly sampled from the source graph. The affinity is conducted by $\mathbf{A}_{ij:ab} = \exp(-|\mathbf{a}_{ij}^S - \mathbf{a}_{ab}^D|^2/\sigma_s^2)$, where $\mathbf{a}_{ij}^S$ measures the Euclidean distance between point $i$ and $j$, and $\sigma_s^2 = 2500$. The edge density is set by $\rho = 1$. One can see when there is no outlier, all methods except for IPFP and SM achieve perfect matching on any gap setting, and we only show the results with outliers. Figure 4 and Figure 3 depict the matching samples and performance curve, respectively. We also show typical converging behavior of GGM$_{\text{lp}}$ and GGM$_{\text{lm}}$ on the upper of Figure 3. We note our path following strategy (Alg. 1) converges slower than multiplicative one (Alg. 2) and they obtain similar final accuracy. One can see when there exist outlier points, GAGM and RRWM suffer notable degraded performance. Our algorithm, on the other hand, achieves competitive performance to state-of-the-arts and behaves stably even under severe degradations.

**Natural Image Matching** This is a challenging dataset as it includes natural images in arbitrary backgrounds. In line with the protocol in [10], 30 pairs of images are included in this test collected from Caltech-101 [37] and MSRC[3]. In each pair of images, MSER detector [38] and SIFT descriptor [39] are used to obtain the key points and the corresponding node feature. Mutual projection error function [40] is further adopted to calculate the edge affinity. The ground-truth are manually labeled. The results are shown in Table 2 and matching examples are shown in Fig. 5. Our method outperforms selected algorithms w.r.t. accuracy regardless of objective score. This also suggests the paradox between accuracy and score under complex affinity modeling as discussed in [35].

# 7 Conclusion

By using Separable Functions, we present a family of continuous approximations to the vanilla QAP formulation widely used in graph matching. We explore the relation of such approximations to the original discrete matching problem, and show convergence properties under mild conditions. Based on the theoretical anslysis, we propose a novel solver GGM, which achieves remarkable performance in both synthetic and real-world image tests. This gives rise to the possibility of solving graph matching with many alternative approximations with different solution paths.

# Acknowledgement

This work was supported in part by a grant from ONR. Junchi Yan is supported in part by NSFC 61602176 and Tencent AI Lab Rhino-Bird Joint Research Program (No. JR201804). Any opinions expressed in this material are those of the authors and do not necessarily reflect the views of ONR.

## Footnotes

[1]Here the number of nodes in two graphs are assumed the same. In case $m \neq n$ one can add dummy nodes as a standard technique as in literature [10, 17].

[2]In fact separable function has its traditional meanings in mathematics, we re-define it in the graph matching context.

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
