[Supplementary Material · supp_material.pdf]

# Supplementary material

All numbered equations with yellow color box such as ( 1 ) are inherited from the main body of manuscript.

## 1 Proof of Theorem 1

**Theorem 1** *The optimal objective $p^*$ to problem ( 2 ) is equal to the optimal objective $p_\delta^*$ to problem ( 4 ).*

**Proof 1** *As problem ( 4 ) is the relaxed version of problem ( 2 ), we must have $p_\delta^* \geq p^*$.*

*Suppose $\mathbf{x}^* = vec(\mathbf{X}^*)$ is the optimal solution to problem ( 4 ). We recursively implement the following procedure until there is no $1$ in $\mathbf{x}^*$. If $\mathbf{x}_{ia}^* = 1$, according to the doubly stochastic property, the $i$th row and $a$th column elements other than $(i, a)$ element would all be $0$. We then remove all the elements in $\mathbf{A}$ corresponding to node $i$ in $\mathcal{G}_1$ and node $a$ in $\mathcal{G}_2$. Finally we can reach a subset of $\mathbf{x}$ and $\mathbf{A}$ such that each element in $\mathbf{x}$ is in the range $[0, 1)$. Figure 1 schematically shows how this procedure works from left to right.*

*However, due to the definition of function $f_\delta$, the affinity score over the remaining nodes becomes $0$. As $\mathbf{A}$ is non-negative, any $1$ value assignment would result in affinity score no less than $0$. Denote the objective value of such assignment $p^{assign}$, then we have $p_\delta^* \leq p^{assign}$. On the other hand, $p^{assign}$ is discrete, then we must have $p^{assign} \leq p^*$.*

*In summary, we have $p^* = p_\delta^*$. QED.*

## 2 Proof of Theorem 2

**Theorem 2** $\lim_{\theta \to 0} p_\theta^* = p_\delta^*$

**Proof 2** *First we define two sets: $\mathcal{C}_1 = \{\mathbf{x} | \mathbf{Hx} = \mathbf{1}, \mathbf{x} \in [0, 1]^{n^2}\}$, $\mathcal{C}_2 = \{\mathbf{x} | \mathbf{x} \in [0, 1]^{n^2}\}$. It's easy to observe that $|p_\theta^* - p_\delta^*| \leq p_1$, where $p_1 = \arg\max_{\mathbf{x}} |\mathbf{h}_\theta^\top \mathbf{A} \mathbf{h}_\theta - \mathbf{h}_\delta^\top \mathbf{A} \mathbf{h}_\delta|$ subject to $\mathcal{C}_1$. This observation is true because the gap between two separable optimal objectives must be no larger than the maximal gap between the objectives.*

*We further define $p_2 = \arg\max_{\mathbf{x}} |\mathbf{h}_\theta^\top \mathbf{A} \mathbf{h}_\theta - \mathbf{h}_\delta^\top \mathbf{A} \mathbf{h}_\delta|$ subject to $\mathcal{C}_2$. As $\mathcal{C}_1 \subset \mathcal{C}_2$, we must have $p_1 \leq p_2$. By rewriting the objective corresponding to $p_2$ in the following way:*

$$\left| \sum_{i,j} \mathbf{A}_{ij} h_\theta(\mathbf{x}_i) h_\theta(\mathbf{x}_j) - \sum_{i,j} \mathbf{A}_{ij} h_\delta(\mathbf{x}_i) h_\delta(\mathbf{x}_j) \right|$$

$$= \left| \sum_{i,j} \mathbf{A}_{ij} \left[ ((h_\theta(\mathbf{x}_i) - h_\delta(\mathbf{x}_i)) h_\theta(\mathbf{x}_j) + (h_\theta(\mathbf{x}_j) - h_\delta(\mathbf{x}_j)) h_\delta(\mathbf{x}_i) \right] \right|$$

*Note $\mathbf{A}$, $h_\theta$ and $h_\delta$ are all bounded. Additionally, $h_\theta(\mathbf{x}_i) \to h_\delta(\mathbf{x}_i)$ and $h_\theta(\mathbf{x}_j) \to h_\delta(\mathbf{x}_j)$ when $\theta \to 0$ by the third property. Thus $|p_\theta^* - p_\delta^*| \leq p_1 \leq p_2 \to 0$. QED.*

Figure 1: Procedure to remove 1 elements. Here the manipulation on a $6 \times 6$ matrix is demonstrated schematically. From left to right, we remove a 1 element and corresponding column and row in each step. The rightmost matrix is $\mathrm{mat}(\mathbf{x}^\dagger)$ with all elements in $[0, 1)$.

## 3 Proof of Proposition 1

**Proposition 1** *For univariate SF $h_{Lap}$, $h_{Poly}$, suppose $p_1^*$ and $p_2^*$ are the optimal objectives for ( 5 ) with $\theta_1$ and $\theta_2$, respectively. Then we have $p_1^* \geq p_2^*$ if $0 < \theta_2 < \theta_1$.*

**Proof 3** *This can be easily proved by showing $h_{Lap}(x; \theta_2) < h_{Lap}(x; \theta_1)$ and $h_{Poly}(x; \theta_2) < h_{Poly}(x; \theta_1)$ when $\theta_2 < \theta_1$. QED.*

## 4 Proof of Theorem 3

**Theorem 3** *Assume that affinity $\mathbf{A}$ is positive definite. If the univariate SF $h_\theta(x) \leq x$ on $[0, 1]$, then the global maxima of problem ( 2 ), which is discrete, must also be the global maxima of problem ( 5 ).*

**Proof 4** *As shown in [1], whenever affinity $\mathbf{A}$ is positive definite, the global maximum of problem ( 3 ) is a permutation. In this case, the optimum to ( 3 ) is also optimum to ( 2 ). Denote $\mathbf{y}^*$ the optimal permutation to ( 3 ). As $\mathbf{y}^*$ is doubly stochastic, it must also satisfy the same constraints in problem ( 5 ). Let $p_1$ be the objective of problem ( 5 ) w.r.t. $\mathbf{y}^*$ – Note $p_1$ is the optimal objective of problem ( 3 ). Assume there exists an optima $\mathbf{x}_\theta^* \neq \mathbf{y}^*$ to problem ( 5 ) with corresponding objective $p_2$. As $p_2$ is optimal, we have $p_2 \geq p_1$. Let $\mathbf{y}_\theta = \mathbf{h}_\theta(\mathbf{x}_\theta^*)$. As $h_\theta(x) \leq x$, we must have $\mathbf{x}_\theta^* \geq \mathbf{y}_\theta \geq \mathbf{0}$. Denote $p_3$ the objective score of ( 3 ) by substituting $\mathbf{x}_\theta^*$. Since $\mathbf{A}$ is non-negative, $\mathbf{x}_\theta^* \geq \mathbf{y}_\theta$ and $\mathbf{x}_\theta^*, \mathbf{y}_\theta \geq \mathbf{0}$, we have $p_3 \geq p_2$. In summary, $p_3 \geq p_1$. However, $p_1$ is the global optimal objective of ( 3 ). Thus the inequality leads to contradiction. The equality exists only when the global optimum of ( 5 ) is $\mathbf{y}^*$. QED.*

## 5 Proof of Proposition 2

**Proposition 2** *Assume affinity $\mathbf{A}$ is positive/negative semi-definite, then as long as the univariate SF $h_\theta$ is convex, the objective of ( 5 ) is convex/concave.*

**Proof 5** *Consider problem ( 5 ), we prove this theorem by checking the property of the Hessian with respect to $\mathbf{x}$. As we have obtained the gradient $2\mathbf{G}\mathbf{A}\mathbf{h}_\theta$ of the objective in ( 5 ) with respect to $\mathbf{x}$, we calculate the Hessian by taking the derivative once again. After some mathematical manipulations, we have $\nabla^2 \mathbf{x} = 2\mathbf{A}\mathbf{K}$, where*

$$
\begin{aligned}
\mathbf{K} = \mathrm{diag}\Bigg( \Bigg[ &\left( \frac{\partial h_\theta}{\partial \mathbf{x}_1} \right)^2 + h_\theta(\mathbf{x}_1) \frac{\partial^2 h_\theta}{\partial \mathbf{x}_1^2}, \\
&..., \left( \frac{\partial h_\theta}{\partial \mathbf{x}_{n^2}} \right)^2 + h_\theta(\mathbf{x}_{n^2}) \frac{\partial^2 h_\theta}{\partial \mathbf{x}_{n^2}^2} \Bigg]^\top \Bigg)
\end{aligned}
\tag{1}
$$

*It is easy to show that $(\partial h_\theta / \partial \mathbf{x}_i)^2$ and $h_\theta(\mathbf{x}_i)$ are non-negative according to Definition 1. As $h_\theta$ is convex, its second order derivative must also be non-negative. Matrix $\mathbf{K}$ is positive semi-definite. Thus the convexity/concavity of $\mathbf{A}$ is preserved after multiplying $\mathbf{K}$. QED.*

# 6 Proof of Proposition 3

**Proposition 3** *Assume affinity matrix $\mathbf{A}$ is positive definite and univariate SF $h_\theta$ is convex. The optimal value to the following problem is:*

$$E_{conv} = \max_{\mathbf{x}} \mathbf{h}_\theta^\top \mathbf{A}^\dagger \mathbf{h}_\theta \tag{2}$$

*Then there exists a permutation $\mathbf{x}^*$, s.t. $E_{conv} - E(\mathbf{x}^*) \leq n\lambda$ where $E(\mathbf{x}^*)$ is the objective value w.r.t. problem ( 5 ).*

**Proof 6** *First for any convex univariate SF $h_\theta$ in range $[0,1]$, we have $h_\theta(x) \leq x$. Under the assumption in the theorem, given $\hat{\mathbf{x}}$ the optima to problem ( 5 ), we can obtain an optimal discrete $\mathbf{y}$ according to the sampling procedure in Theorem 1. The optimal objective of ( 5 ) can be written as:*

$$
\begin{aligned}
E_{conv}(\mathbf{y}) = \sum_{i \neq j, a \neq b} & \mathbf{A}_{ij:ab} h_\theta(\mathbf{y}_{ia}) h_\theta(\mathbf{y}_{jb}) + \\
& \sum_{i,a} \left( \mathbf{A}_{ii:aa} + \lambda \right) h_\theta^2(\mathbf{y}_{ia})
\end{aligned}
\tag{3}
$$

*Besides, by substituting $\mathbf{y}$ into problem ( 5 ) we obtain:*

$$E(\mathbf{y}) = \sum_{i,j,a,b} \mathbf{A}_{ij:ab} h_\theta(\mathbf{y}_{ia}) h_\theta(\mathbf{y}_{jb}) \tag{4}$$

*By subtracting Equation (4) from (3) we have:*

$$E_{conv}(\mathbf{y}) - E(\mathbf{y}) = \lambda \sum_{i,a} h_\theta^2(\mathbf{y}_{ia}) \tag{5}$$

*As $\mathrm{mat}(\mathbf{y}) \in \{0,1\}^{n^2}$ is a permutation hence $h_\theta(\mathbf{y}_{ia}) = \mathbf{y}_{ia}$, we have $\lambda \sum_{i,a} h_\theta^2(\mathbf{y}_{ia}) = n\lambda$. Then there exists at least one permutation $\mathbf{x}^*$ satisfying the condition. QED.*

## References

[1] A. Yuille and J. Kosowsky, "Statistical physics algorithms that converge," *Neural Computation*, vol. 6, pp. 341–356, 1994.