[Reviews · NeurIPS 2018]

Reviewer 1



Summary: This paper shows that a large family of functions, defined as Separable Functions, can asymptotically approximate the discrete matching problem by varying the approximation controlling parameter. They present Theorem 1 that The optimal objective p^* to problem (2) is equal to the optimal objective p_delta^* to problem (4), in which problem (2) is original graph matching problem in integer domain and problem (4) is the problem relaxed to continuous domain. They define Separable Function (SF), f_theta(x, y), that satisfies there properties mentioned in Definition 1 in Section 4.1, and theta in univariate SF h_theta is defined as controlling parameter. The SF has three fine properties for computation: 1) it shows similar behavior as a probabilistic distribution on two independent variables; 2) it eases gradient computing; 3) we can construct new approximation function via reweighted summation and multiplication of existing Separable Functions. They show that original problem (4) can be asymptotically approximated using SFs as theta -> 0. They analyze the properties of convexity/concavity under such approximations, as there is technique to convert non-convex/non-concave problems into convex/concave ones. They describe two optimization strategies for generalized graph matching (GGM): Path Following Strategy and Multiplication Strategy. The first one starts by obtaining a local optimum from a relatively tractable problem , then we shrink the value of theta by a factor of two. For the second one, each step amounts to a multiplication x(t+1) = Ax(t) and the objective score over the solution path is non-decreasing, under the assumption that A is positive semi-definite. It is found the multiplicative method converges much faster and hence the overall run time is less compared with the path following method. They perform experiments on two popular random graph matching datasets, CMU house sequence and Caltech-101/MSRC object matching, with comparison to Spectral Matching (SM), Integer Projected Fixed Point (IPFP), Graduated Assignment (GAGM), Reweighted Random Walk (RRWM), Soft-restricted Graduated Assignment (SRGA), Factorized Graph Matching (FGM), and Branching Path Following Matching (BPM). The experiment shows their GMM methods reach similar performance or outperform other state-of-the-art methods. Strengths: - This paper is technically sound with experiment results to support the effectiveness of their approach. - This paper provides novelty algorithms for GMM problems with strong theoretical analysis. - The theoretical analysis and experiments in this paper is written well and clear. Weaknesses: - It may be better to provide asymptotic analysis of running time for both proposed GMM optimization algorithms. Rebuttal Response: After reading the rebuttal, the author addresses some comments on computation complexity, convergence criterion, parameter sensitivity, and others. I support accepting the paper, but I will not change my score.

Reviewer 2



The paper introduces a novel optimization strategy for solving the graph matching problem. It starts with the original combinational optimization formulation and proposed a couple of relaxations, including, relaxing the integer variables to real variables, approximating the quadratic objective function using separable functions, which are then relaxed using a series of smoother functions (where the limit of the smoother functions approximate the separable function). The final objective function is optimized in two ways, namely, using path-following strategy and multiplication strategy. I like reading the paper. The writing is clear. When introducing a relaxation, the outcome and various properties are explained. The idea of using separate functions is interesting and appears to be useful. On the down side, I would like to see more challenging examples. Also, what is the key benefit of this relaxation? Does it merely improve the quality of the matching, or the relaxation makes it more scalable to large-scale GM problems. The improvements in map quality are not salient. So if it is the latter case, it would be good to share more examples. Overall, the submission is solid. I am positively inclined, and if other reviewers are positive, then I am support accepting the paper. Minor comments: 1. What is the converging criterion for Algorithm 1 and Algorithm 2, respectively?

Reviewer 3



The authors propose a novel relaxation approach for the graph matching (GM) problem under a quadratic assignment problem formulation. They relax the assignment using a continuous formulation and then factorize it via a product of what they call "separable functions". These functions are parametrized by \theta with guarantees that in the limit of theta approaching 0 the solution to the original optimization problem is recovered. The authors the propose two optimization strategies one of which involves iteratively restarting from solutions found decreasing theta. The main drawback of the work is a lack of defense for its merits: it is empirically shown that the approach compares well to state of the art (SOTA) alternatives, but it is not emphasized what are the characteristics that make the approach relevant: is it more efficient? the author hints at this in the caption of a figure where they report that SOTA is one order of magnitude slower; there should be more emphasis on the complexity aspects. Are there classes of graph matching problems that are difficult for some of the competitive approaches that become then better tractable by the proposed approach? The authors use as benchmarks random graphs where they show comparable performances to other SOTAs but do not indicate any advantage of the proposed approach. A sensitivity analysis w.r.t. the few parameters of the method is also lacking (how is the quality of the solution affected by changing theta, alpha and kappa?). However, disregarding some minor issues with the level of the English language that is overall acceptable, the approach is of interest and worth being presented as it leaves interesting open questions on the best way to solve the new optimization formulation for the GM problem. After rebuttal: The authors have addressed the computational complexity concern and have justified the merit in terms of how flexible the method is. I support accepting the paper, but would not change the score.